# Comparative Study on Macro-Tribological Properties of PLL-g-PEG and PSPMA Polymer Brushes

**DOI:** 10.3390/polym14091917

**Published:** 2022-05-08

**Authors:** Fue Ren, Shuyan Yang, Yang Wu, Feng Guo, Feng Zhou

**Affiliations:** 1School of Mechanical & Automatic Engineering, Qingdao University of Technology, Qingdao 266520, China; renfue@qut.edu.cn (F.R.); mefguo@qut.edu.cn (F.G.); 2State Key Laboratory of Solid Lubrication, Lanzhou Institute of Chemical Physics, Chinese Academy of Sciences, Lanzhou 730000, China; yangwu@licp.cas.cn

**Keywords:** PLL-g-PEG brushes, PSPMA brushes, friction, film formation property, hydration effect, hydrodynamic effect, lubrication mechanism

## Abstract

An ultra-low friction and high load capacity could be obtained on the surfaces grafted by a polymer brush even at relatively slow friction speeds in aqueous lubrication environments, which has attracted widespread attention to study the lubrication mechanism of polymer brushes; however, it has yet to be fully understood. The macroscopic tribological performance of two different polymer brushes, which were prepared by the method of “grafting to” and “grafting from” and named as PLL-g-PEG and PSPMA, respectively, were investigated. The friction results demonstrated that PLL-g-PEG obtained a lower friction coefficient than polymer brush PSPMA, which was ascribed to its unique “self-healing” behavior. The lubrication film was in situ observed and the film thickness induced by the polymer brush was measured using a laboratory set for film thickness measurement apparatus based on interference technology. It was found that PSPMA exhibited excellent lubrication performance not found in PLL-g-PEG, and two film-forming mechanisms highly dependent on velocity were revealed, which may be important to interpret the lubrication mechanism of polymer brushes in aqueous lubricants.

## 1. Introduction

Polymer brushes are thin films of polymer chains covalently anchored to surfaces, and was found to enhance the lubrication properties of water with ultra-low friction [1], such as hip joint lubrication, and polymer brushes grafted to the rubbing surface could effectively reduce the friction and wear due to the hydration layer of the brush adsorbed on the substrate [2,3,4,5]. There are generally two methods for preparing polymer brushes, one of which is “grafting to the surface” (hereinafter referred to as “grafting to”), and the other which is “grafting from the surface” (hereinafter referred to as “grafting from”).

A polymer brush is grafted on the substrate by physical adsorption [6] or chemical bonding [7] in the preparation method of “grafting to”. It was found that a polymer brush, such as poly (L-lysine)-g-polyethylene glycol (hereinafter referred to as PLL-g-PEG), could be continuously re-grafted on the rubbing surface even though it was continuously destroyed by shear stress during the rubbing process, forming a unique “self-healing” property for achieving low friction [8,9]. However, since the maximum bonding density of polymer chains decreases with the increase in polymer molecular weight, the load-bearing capacity of polymer brushes is also limited [10]. A polymer brush is achieved by surface-initiated polymerization in the preparation method of “grafting from” [11]. Firstly, a suitable polymerization initiator is anchored to the surface of the substrate (the initiator used can be ionic or free radical), then the monomer is diffused into the initiator to produce a polymer brush through covalent bonding. Surface-initiated atom transfer radical polymerization (SI-ATRP) is viewed as an excellent polymerization technology that has developed rapidly in recent years for its various structures on many substrates made of different materials, such as silicon, glass, etc. [12]. It should be noted that polymer brushes cannot be regenerated once sheared from the surface due to the limit of this preparation method.

Inspired by the microstructure and lubrication mechanism of natural synovial fluid joints, scientists found that even at relatively slow friction speeds (10^−2^~10^−1^ m/s), an extremely thin lubricating film could be formed on the surfaces grafted by a polymer brush, thus obtaining an ultra-low friction coefficient (10^−2^~10^−3^). Moreover, the friction coefficient does not depend on the sliding speed [8,13,14]. This interesting phenomenon has attracted great attention of scientists, and for the first time, the polymer brush was introduced in the tribology research field by Klein [15], where it was found that the normal force between the surfaces of two compressed mica surfaces grafted by a polymer layer in toluene increased significantly when the surface moved in parallel, and the friction force decreased significantly even at low sliding speeds. Since then, a series of research studies have been conducted to study tribological properties of polymer brushes. For example, Spencer et al. [16,17,18] demonstrated that an extremely low friction coefficient can be achieved when added by the PLL-g-PEG polymer into the HEPES lubricant conducted on a pin-disk friction tester, and proposed that the fast “self-healing” behavior of PLL-g-PEG is mainly attributed to the reduction in friction and wear. Zhou et al. [19] found that the polymer brush as coating exhibited good properties in pure water and different biological media by surface-initiated polymerization (SIP), and its friction coefficient could be as low as 0.015, which could have a wide range of applications in biological implants and medical devices. Wei Qiangbing et al. [20] successfully achieved the adjustment control of friction by preparing a responsive polyelectrolyte brush with polymethacryloyloxyethyl dimethyl ammonium chloride, which could change the good lubricating condition (μ~10^−3^) to an ultra-high friction condition (μ > 1). Qu et al. [21] found that the hairy NP polymer as an additive in PAO exhibited excellent lubricating properties, which could effectively reduce the friction and wear conducted on the PlintTE-77 friction tester. Takahara et al. [22] firstly measured the gap between the ball and disk of a point contact test apparatus using optical interferometry technology, where the glass disk was grafted by polymer, and a fluid lubricating layer was formed assisted by a polymer brush on contact, while it was not observed for bare glass.

Since the polymer brush has been successfully applied in urinary catheters as coating due to its good lubrication performance, it has greatly eased the pain of the patients. Additionally, it can be view as potential green lubrication for its reduction in friction and wear. Regrettably, the water lubrication mechanism of polymer brushes has not been understood so far, and consequently, their wide application has been limited, but it is agreed that the hydration effect of polymer brushes is the key factor. The tribological properties, such as friction and wear, have been intensively investigated, while few studies have been conducted to study the film formation and lubrication mechanism of polymer brushes considering hydrodynamics. Based on this, this study comprehensively studied the effects of polymer brushes (PLL-g-PEG and PSPMA) grafted by different preparation methods on the macro-tribological properties under an aqueous lubrication environment by measuring friction. Then, we investigated their lubrication behavior through in situ observation, and the film thickness induced by the polymer brush was measured on a laboratory set for film thickness measurement apparatus based on interference technology. The film-forming mechanism of aqueous lubrication was further revealed, which provided necessary technical support for guiding the design of polymer lubricating materials, and which was important to further explore the lubrication mechanism of the polymer brush in an aqueous lubricant and develop green lubrication.

## 2. Materials and Methods

### 2.1. Test Materials of PLL-g-PEG Brushes

The materials contained 30% hydrogen peroxide, 98% concentrated sulfuric acid, sodium hydroxide, N-(2-hydroxyethyl) piperazine-N-(-2-ethanesulfonic acid) solvent (HEPES solvent, purchased from Shanghai Yuanye Biotechnology Co., Ltd., Shanghai, China), and poly(L-Lysine)-g-poly(ethylene glycol) (PLL(20)-g(3.6)-PEG(5) solvent (purchased from Creative PEGworks Company, Durham, NC, USA)). The specific copolymer used, PLL(20)-g(3.6)-PEG(5), has a PLL molecular weight of 20 kDa, a PEG side chain molecular weight of 5 kDa, and a lysine to PEG side chain graft ratio of 3.6. The chemical reagents used in the experiment were all of analytical grade.

### 2.2. Test Materials of PSPMA Brushes

The materials: Dopamine hydrochloride (98%), tris(hydroxymethyl)aminomethane (THAM, 99.8%), and potassium 3-sulfonic acid propyl methacrylate (SPMA, 96%) (purchased from Aladdin Reagent (Shanghai) Co., Ltd., Shanghai, China); 2-Bromo-2-methylpropionyl bromide (98%) (purchased from Beijing Bailingwei Technology Co., Ltd., Beijing, China); cuprous bromide (CuBr) and sodium tetraborate (Na_2_B_4_O_7_) (purchased from Sinopharm Chemical Reagent Co., Ltd., Shanghai, China); and methanol, ethanol (99%), dichloromethane, 2,2’-bipyridine (bipy), copper chloride (CuCl_2_), and magnesium sulfate (MgSO_4_) (purchased from Tianjin Bodi Chemical Co., Ltd., Tianjin, China). The reagents were analytically pure and had not undergone further processing before use. The deionized water used was self-made in the laboratory.

### 2.3. Preparation of Polymer Brush PLL-g-PEG

First, 0.238 g of the HEPES solvent was dissolved in about 90 mL deionized water, diluted to 100 mL with deionized water, and the pH was adjusted to 7.4 with 1 mol/L NaOH to obtain a HEPES solution with a concentration of 10 mM. Then, we weighed the PLL-g-PEG solvent and dissolved it in a 10 mM HEPES solution, and prepared a HEPES aqueous solution with a concentration of 0.25 mg/mL of PLL-g-PEG according to the required amount of the experiment. The lubricants used in the test were the HEPES solution and the HEPES aqueous solution of PLL-g-PEG with a concentration of 0.25 mg/mL. The molecular structure of -PEG is shown in Figure 1a.

The surface of the substrate was ultrasonically cleaned with petroleum ether, absolute ethanol, and deionized water, dried with nitrogen, then immersed in a Pirahan (H_2_SO_4_:H_2_O_2_ = 70:30, *V*/*V*) solution at 90 °C for 30 min. After repeated rinsing with a large amount of deionized water, it was dried with nitrogen again, then irradiated with ultraviolet for 20 min by a vacuum ultra-ultraviolet surface treatment device, thereby obtaining the surface of the silicon substrate rich in hydroxylation on the surface. The hydroxylated surface was fully immersed in the aqueous solution prepared above. Since the polymer backbone PLL has a positive charge (NH_3_^+^) in a neutral environment, it attracted each other on the hydroxylated surface with a negative charge (OH^−^) and was grafted to the substrate surface. The hydroxylated surface was fully immersed in the aqueous solution prepared above. Since the polymer backbone PLL has a positive charge (NH_3_^+^) in a neutral environment, it attracted each other on the hydroxylated surface with a negative charge (OH^−^) and was grafted onto the substrate surface. The polymer side chain PEG is easy to bind a large number of water molecules, forming a nano-thick polymer brush in a neutral environment to form a hydration layer with a good lubricating effect. During the experiment, the soaking time was 30 min to ensure sufficient grafting of the polymer brush on the surface of the substrate. The schematic diagram of the adsorption of polymer PLL-g-PEG to the surface of hydroxylated substrates in an aqueous environment is shown in Figure 1b.

### 2.4. Preparation of Polymer Brush PSPMA

Using the SI-ARTP technology to graft the polymer brush PSPMA on the surface of the steel ball and the glass disk was mainly achieved through the assembly of the surface initiator and the initiation of polymerization. The preparation method is shown in Figure 1c. Before grafting, the dopamine concentrate (DOPAMA) needed to be prepared first. For the specific synthesis process, please refer to the method reported in reference [23]. The substrates were the surfaces of steel balls and glass disks. Before preparation, the steel balls and glass disks were immersed in petroleum ether for ultrasonic treatment for 30 min. Then, they were rinsed with ethanol, rinsed with deionized water, and finally blow-dried with nitrogen to remove surface impurities.

A certain amount of the prepared dopamine concentrate was added to ethanol and stirred at room temperature for 10 min. The steel ball was completely immersed in this solution for the assembly of the substrate surface initiator, and stored in the dark for 12 h. Then, we removed the steel ball and rinsed the surface of the substrate with a large amount of ethanol. A certain amount of monomer 3-sulfonic acid propyl methacrylate potassium salt (SPMA) was added to the mixed solution of methanol and water with a volume ratio of 1:2, and nitrogen protection was introduced and stirred at room temperature for 30 min. We added catalyst 2,2′−bipyridine (bipy) and cuprous bromide (CuBr) to it in turn, and continued stirring for 20 min under nitrogen protection until a homogeneous reddish-brown monomer solution was formed. We placed the clean steel balls or glass disks that had been assembled with the initiators into the monomer solution, removed them after 4 h of surface-initiated polymerization under nitrogen protection, then rinsed the substrate surface with a large amount of deionized water. Lastly, we blow-dried with nitrogen for use.

Due to the large size of the grafted surface, the preparation method of the glass disk surface was slightly complicated, and the assembly of the surface initiator needed to be completed in two steps. In the first step, an appropriate amount of tris buffer solution with a concentration of 10 mmol/L was prepared, and a dopamine hydrochloride solution with a concentration of 0.5 mg/mL was prepared with it. The glass plate was fully soaked in the solution for 48 h, then the glass plate was removed and rinsed with deionized water. In the second step, the glass plate was soaked in a mixed solution of 2-bromo-2-methylpropionyl bromide and dichloromethane with a volume ratio of 1:200 for 12 h, then the glass plate was removed and the surface was rinsed with ethanol. The monomer solution used in the preparation of surface-initiated polymerization was similar to the steel ball method. The difference was that the glass surface to be grafted needed to be placed on a polished zinc plate for polymerization reaction. A tiny gap was left, and the prepared monomer solution was dropped into the gap between the zinc plate and the glass plate, after which the monomer solution filled the entire gap by capillary force, and the glass plate was removed after 3 h of polymerization. The substrate surface was rinsed with plenty of deionized water, and finally blow-dried with nitrogen for use.

### 2.5. Test Rigs and Measurement Methods

Figure 1d is a schematic diagram of the friction tester consisting of a steel ball and 3 glass plates fixed on Anton Paar’s Physica MCR rheometer (Anton Paar Co. Ltd., Graz, Austria). The ball measurement fixture included a connecting shaft, a test ball, and a fixing device, and the 3 glass plates were fixed on a specially designed spring system to ensure that the applied force of the three plates was uniform. The diameter of the ball was 12.7 mm, the size of the plate was 15 mm × 6 mm × 3 mm (L × W × H), and they were made of GCr15 steel and K9 glass, respectively. Glass plates (Ra = 0.006 μm) were purchased from Wuxi Delmon Technology Co., Ltd., and the steel ball (Ra = 0.040 μm) was purchased from Qingdao Meike Precision Machinery Co., Ltd. Before the friction test, sufficient lubricant of about 4 mL was added to the contact area to fully submerge the ball and 3 plates. For polymer brush PLL-g-PEG, the steel ball and glass plates needed to be immersed in the mixture of HEPES and PLL-g-PEG with 0.25 mg/mL concentration followed by a wait of 30 min to ensure that their surface was adequately grafted by polymer brush PLL-g-PEG, for it was found that the contact angle did not change after 30 min during the subsequent contact angle measurement (seen in Appendix A), which suggested that the PEG polymer brush grafted may have reached saturation and was completely grafted on the substrates. For polymer brush PSPMA, the ball and plates also needed to be completely soaked in deionized water for 5 min to ensure that the PSPMA brush grafted on the substrates could fully swollen and stretch, which was consistent with the change in contact angle with time in Figure 2b.

The loads applied to the ball were 0.5 N and 3.5 N (corresponding to the maximum Hertz contact stress, P_Hertz_ was 217 MPa and 416 MPa), and all friction coefficients were measured starting from a velocity Ue of 1.4 m/s and successively decreasing to 0.003 m/s, which could basically cover the range from fluid dynamics to the boundary lubrication regime. The ball–plate friction pairs had to be replaced with fresh pairs for each test. Each experiment was repeated 3 times and the average value was taken.

The laboratory set for observing the contact area in situ and measuring film thickness by the optical interference method is shown in Figure 1e. The diameter of the glass disk was 150 mm, which was made of K9 glass coated with approximately 20 nm of chromium as well as 200 nm of silica to increase its reflectivity and purchased from Shanghai Weipu Optoelectronics Technology Co., Ltd., Ra = 0.005 μm; the diameter of the steel ball was 25.4 mm and it was made of GCr15 (purchased from NSK Company, Tokyo, Japan, Ra = 0.043 μm). Two lasers, with wavelengths of 655 nm and 532 nm, were shone through the glass disk onto the contact area at the same time. The interference fringes produced due to optical phase differences of reflected light from the glass disk and the steel ball were captured via a CCD camera. Maps of lubricant film thickness in the contact area, with a thickness resolution of 1 nm, were obtained [24].

Before the test, the ball–disk friction pair was ultrasonically cleaned with petroleum ether for more than 30 min, then the petroleum ether was cleaned with absolute ethanol, dried with nitrogen, and fixed in the specified position of the testing machine for the test.

The JC2000C1B contact angle meter was used to characterize the change in the wettability of the substrate (steel or glass) due to the grafted polymer brush, which was purchased from Shanghai Zhongchen Digital Technology Equipment Co., Ltd.

The experimental temperature was controlled at 22 ± 1 °C, and the humidity was kept at RH50 ± 2%.

## 3. Results and Discussion

### 3.1. Influence of Polymer Brush on Wetting Properties of Substrate

It was confirmed that wettability plays an important role on tribological performance [14], which is an important indicator for analyzing the lubrication performance of solid-liquid surfaces. In the study, the contact angle was used to measure the wettability. The contact angle before and after grafting were measured by the contact angle measuring instrument JC2000C1B, and the change in substrate wetting performance after grafting polymer brushes was qualitatively characterized; the wettability of the surface of the friction pair played an important role in its lubricating effect. It was an important indicator for analyzing the lubricating characteristics of the solid-liquid surface. The change in contact angles of glass and steel substrates lubricated by HEPES added PLL-g-PEG of 0.25 mg/mL concentration are shown in Figure 2a.The graph shows a decrease with time in the contact angle by adding the polymer PLL-g-PEG to the HEPES solution, which no longer changed and maintained 5.5° or 45° after 1800 s, indicating that the polymer PLL-g-PEG continuously grafted to the substrates and was finished at about 1800 s. Additionally, the wettability of substrates was improved due to the PEG side chain by binding the surrounding water molecules. For polymer brush PSPMA, there was a small change with time in the contact angle between the glass and steel substrate as shown in Figure 2b. This was because PSPMA had been successfully grafted from substrates by initiate polymerization and the contact angle was much lower compared with the bare substrate, meaning that the polymer brush PSPMA could also improve the wettability of the substrate, but PSPMA was more effective in enhancing wettability especially for the steel substrate.

### 3.2. Friction Comparison of Polymer Brush PLL-g-PEG and PSPMA

The Stribeck curves were used to study the water lubrication mechanism of polymer brushes from full-film hydrodynamics to boundary lubrication regimes, as shown in Figure 3. For 0.5 N, as shown in Figure 4a, the Stribeck curves of the two polymer brushes were relatively close, presenting a fluctuation in lower velocity and a sharp decrement to an extremely low friction coefficient (0.018 and 0.028) without a change with velocity, which may be ascribed to the hydration layer on the rubbing surface induced by polymer brushes. However, there was some difference in the Stribeck curves in Figure 3b; for example, the friction coefficient of PLL-g-PEG was much lower than that of PSPMA, which meant that PLL-g-PEG was more effective in reducing friction because PSPMA grafted on the ball and plates could not regenerate new polymer brushes after they were destroyed at higher loads. Moreover, the fluctuation in the Stribeck curve of PLL-g-PEG was not observed at an extremely low velocity, which suggested that a stable hydration layer always existed in contact.

It was found in Figure 3 that the reduction in friction was affected by the applied load and the entrainment velocity U_e_, which corresponded to the compressive stress and shear stress experienced by the polymer brushes, respectively. For studying how they influence the friction and the effects on friction, the friction coefficient curve with time was compared at different loads and velocities as shown in Figure 4. The friction coefficient of the polymer brush PLL-g-PEG was much lower than that of the polymer brush PSPMA at the load of 0.5 N, the friction coefficient was as low at about 0.003 especially at the entrainment speed of 300 mm/s, and it could be considered that super lubrication had occurred in this case, indicating that PLL-g-PEG exhibited excellent anti-friction performance that PSPMA did not have. It was also found that the friction coefficient was much higher at a velocity of 10 mm/s, which could be explained by the fact that the hydrodynamic effect is enhanced at higher velocities so that there was always a stable fluid film in the contact area even under higher shear stress. At a higher load of 3.5 N, both PLL-g-PEG and PSPMA polymer brushes had poor performance in reducing friction compared with the low load, mainly due to the collapse of the “stretched” polymer brush at higher compressive stress, resulting in a larger area of solid–solid direct contact and a higher friction coefficient.

### 3.3. Anti-Friction Mechanism of Polymer Brush PLL-g-PEG

What made the PLL-g-PEG polymer brush maintain an extremely low friction force in the 2 h rubbing test as shown in Figure 4? Several sets of experiments were conducted at the load of 0.5 N for exploring the anti-friction mechanism of PLL-g-PEG seen in Figure 5a–d. Here, the black curve was obtained for the bare steel ball and glass plate lubricated by pure HEPES on the friction tester for 2 h, the green curve was obtained for the steel ball and glass plate which was fully grafted by PLL-g-PEG lubricated by the HEPES aqueous solution added by PLL-g-PEG with a concentration of 0.25 mg/mL. The blue curve was obtained by fully grafting the polymer brush (called pre-coating) to the ball and glass surfaces in advance, in which the lubricant used was pure HEPES. By comparing these three different curves in Figure 5a,b, it could be seen that after the surface was pre-grafted by the polymer brush represented by the blue curve, the friction coefficient only decreased for a short time and then was close to that of the bare surface (seen in the black curve), which was much higher compared with the green friction curve, and it could be ascribed to the fact that the polymer brush was destroyed at the rubbing process and could not be regenerated lubricated by pure HEPES.

The composite curve of the black and orange curves demonstrates the change in the friction coefficient with time, and among them, the black curve was obtained for ungrafting the ball and plates lubricated by pure HEPES with the volume of 4 mL within 1000 s. At the time of 1000 s, the friction test was stopped, and consequently, 2 mL of pure HEPES solution was removed from the oil pool with a pipette, then the PLL-g-PEG polymer with a concentration of 0.5 g/mL was injected successively, when an HEPES aqueous solution with a concentration of 0.25 mg/mL was obtained after dilution. The glass plate and ball was immersed into this solution for 30 min for grafting the full polymer. The friction coefficient decreased rapidly after the injection of the PLL-g-PEG polymer (the orange curve seen in Figure 5c,d), and we finally achieved an extremely low friction coefficient, which was close to the green curve, illustrating that PLL-g-PEG had the property of rapid “self-healing”, i.e., it was continuously destroyed due to high shear stress during the rubbing process while it simultaneously could be rapidly re-grafted to the rubbing surface under the action of electrostatic force, and the two reached a dynamic equilibrium of adsorption–desorption–resorption after a short period of time. This self-healing behavior effectively reduced the friction. The “self-healing” behavior of PLL-g-PEG is illustrated in Figure 6.

### 3.4. Film-Forming Properties: Comparison of Polymer Brush PLL-g-PEG and PSPMA

A laboratory set for film thickness measurement device was used to observe and measure the lubrication film in the contact area composed of a steel ball and glass disk grafted by the polymer brush PLL-g-PEG or PSPMA based on optical interferometry technology, and the film-forming properties of these two polymer brushes were investigated. Firstly, 10 mL of lubricant (HEPES added by PLL-g-PEG, deionized water for PSPMA) was injected into the oil pool. When the ball rotated, the lubricating oil could be continuously involved in the contact area, thus ensuring adequate lubricating oil supply. Keeping the steel ball and glass disk running at same speed of 16 mm/s for 30 min without applying load, it could drive the lubricant into the contact area to ensure the steel ball and glass disk were fully grafted by the polymer brush and fully lubricated. Secondly, a load of 4 N was applied to the ball for which the corresponding Herze contact pressure was about 274 MPa, and the ball and glass disk could roll and rotate at the same speed controlled by their respective servo motors, and a pure rolling of the slip-rolling ratio (abbreviated as SRR) of 0 meant there was no shear stress. The entrainment speed gradually increased from 1 to 512 mm/s without stopping, simultaneously capturing interference images in contact at each speed using CCD as shown in Figure 7a. Each lubricant test had to be replaced with a fresh ball–disk friction pair, each group of experiments was repeated five times, and the average of film thickness was measured. Based on these images captured, the film thickness could be analyzed to be calculated, and the measurement principle can be found in reference [25].

For the polymer brush PLL-g-PEG, there was a slight change in optical interference in the contact area with the increase in entrainment speed, indicating that film thickness variation was small, and scratches on the glass disk began to appear at the entrainment speed of 96 mm/s, which meant that an effective lubricating film no longer existed in the contact area, and accordingly, the test was forced to stop. Surprisingly, the surface of the steel ball and glass disk modified by the PSPMA brush showed no scratches throughout the experiment. The optical interferences of PSPMA polymer brushes presented big changes with increasing entrainment speeds, such that at a lower speed the image appeared dark, while the color became brighter at several speeds greater than 64 mm/s. Here, the change in color represented the change in light intensity in the contact area, and in turn represented the change in lubrication film thickness; the color changed from brown to yellow-green at lower speeds and the changed to yellow-red at higher speeds, suggesting that the load capacity of the lubrication film was high enough to separate the ball from the glass disk at this time and that it exhibited excellent lubricating performance. It was also observed that the shape of the contact area changed from a circle to an ellipse as the speed increased.

The central film thickness in the contact could be calculated according to the light intensity change recorded from the optical interferences in Figure 7a, and the curves of film thickness vs. entrainment speed are shown in Figure 7b. It could be found that the maximum film thickness achieved by the PLL-g-PEG brush was about 37 nm at the speed of 64 mm/s, which demonstrated that the PLL-g-PEG brush had poor film formation and load capacity compared with the PSPMA brush. It was speculated that the PEG side chain was bound to the surrounding water molecules, and the bound water and free water in the polymer chain were rapidly exchanged to form a thin hydration layer, while the load capacity was weak limited by the graft density and thickness of the polymer brush.

For the PSPMA brush, at low entrainment speeds (below 16 mm/s), the film thickness was about 35 nm without changes in the independence of entrainment speeds, and when the entrainment speed was greater than 32 mm/s, the film thickness suddenly increased, and grown linearly with speed in the logarithmic coordinate, and the water film thickness increased up to 100 nm at a speed of 128 mm/s. It could be concluded that PSPMA brushes had excellent film-forming properties and a greater load-carrying capacity.

### 3.5. Lubrication Mechanism of Polymer Brush PSPMA

#### 3.5.1. Influence of Load on Film Formation

The friction results confirmed that polymer brushes may be collapsed at a higher load. Keeping the load increase from 2 N to 32 N while other test conditions remained unchanged, the effect of load on the film-forming properties of polymer brushes was explored, and the related measurement results are shown in Figure 8a–d. There was almost no change in the optical interference image. The film thickness at lower loads of 2 N and 8 N, was close to that obtained at 4 N as shown in Figure 8e. For higher loads (16 N and 32 N), the film thickness was also close to that of lower speeds; however, the film thickness decreased suddenly when the entrainment speed was over 64 mm/s and even decreased with increasing speed at 32 N because when compressive stress was relatively high, the polymer brush suffered to collapse and partially failed.

#### 3.5.2. Influence of SRR on Film Formation

Except for the impact of compressive stress on the polymer brush, the shear stress also has an impact on the film-forming properties of the polymer brush, for excessive shear stress can lead to polymer brush peeled from the substrate, which can weaken or even fail the lubrication effect. In this study, shear stress could be changed by changing SRR. To increase SRR from 0 to 0.7 while other test conditions remained unchanged, the effect of SRR on the film-forming properties of polymer brushes was studied at the load of 4 N, and the related measurement results are shown in Figure 9. At lower SRRs (0.05, 0.1, and 0.3), the optical interference images were similar to that at SRR = 0, and with the increase in SRR, the shape of the contact area was basically circular (SRR = 0.5, 0.7), which turned into an ellipse at a higher speed and lower SRR.

The film thickness at the lower SRR of 0.05, 0.1, and 0.3 was close to that obtained at SRR = 0 as shown in Figure 9f. For higher SRR (0.5 and 0.7), the film thickness was also close to that obtained at a lower speed; however, the film thickness decreased suddenly when the entrainment speed was over 256 mm/s and even decreased with increasing speed, which was due to the fact that when shear stress was relatively high, the polymer brush was partially sheared off from the substrate and failed.

#### 3.5.3. Lubrication Mechanism of Polymer Brush PSPMA under the Combined Effect of Hydrodynamics and Hydration

The average central film thickness curve shown in Figure 10a shows that the film thickness of the lubricating film in the contact area strongly depended on the entrainment speed, presenting two different lubrication mechanisms: it was considered to be in the thin-film lubrication regime at lower speeds, where film thickness was independent of velocity and was stable at 30~40 nm, and the hydration effect of polymer brush was responsible for the establishment of an effective film in contact; when velocity was greater than 32 mm/s, it transformed into elastic hydrodynamic lubrication, for the film thickness and speed increased almost linearly in a logarithmic coordinate system.

Figure 10b,c are the film thickness profiles along the X and Y directions in the contact area, respectively. It was found that the shape of the central film thickness changed; that is, it was flat at low entrainment speeds (below 16 mm/s), indicating that the end leakage was small, which is one of the characteristics of thin-film lubrication that distinguishes it from elastohydrodynamic lubrication [26]. When the speed was greater than 32 mm/s, it was found that the film thickness curve along the X direction became more and more curved with the increase in speed, while the film thickness curve along the Y direction was no longer a straight line but an oblique distribution, which meant that the hydrodynamic effect on lubricating film became stronger and stronger; simultaneously, obvious cavitation (cavitation, as shown in Figure 10c) was observed in the exit, which also confirmed the fluid film induced by hydrodynamics [27]. It was also found that the measured value of film thickness was higher than that of film thickness predicted by the Hook film thickness formula (2~12 nm) added by film thickness due to hydration effect (~35 nm), when it is regarded as isoviscous-elastic lubrication (i-EHL) [28]. It was proposed that the polymer brush presented surprising lubrication enhancement resulting from the synergistic effect of the hydration effect and hydrodynamic effect.

## 4. Conclusions

(1) Two polymer brushes were prepared on the surfaces of rubbing pairs (ball-on-3 plates and ball-on-disk) by preparation methods of “grafted to” and “grafted from”, referred to as PLL-g-PEG and PSPMA, respectively, and their macro-tribology properties including friction and lubrication film thickness were compared in an aqueous lubrication environment, where it was found that PLL-g-PEG had better performance in reducing friction, while PSPMA had better lubrication performance with high load capacity.

(2) It was confirmed that PLL-g-PEG has the unique property of rapid “self-healing”, which means that the polymer brush was continuously destroyed due to high shear stress and comprehensive stress during the rubbing process, while simultaneously, it could be rapidly re-grafted to the rubbing surface under the action of electrostatic force, and a dynamic equilibrium of adsorption–desorption–resorption was reached after a short period of time. It was this “self-healing” behavior that effectively reduced friction.

(3) Optical measurement results presented two different lubrication film formation mechanisms that strongly depended on the entrainment speed: it was considered to be in the thin-film lubrication regime at lower speeds (<16 mm/s), where film thickness was independent of velocity and was stable at 30~40 nm induced by the hydration effect of the polymer brush rather than the hydrodynamic effect; when velocity was greater than 32 mm/s, the film thickness and speed increased almost linearly in a logarithmic coordinate system, and the polymer brush exhibited surprising lubrication enhancement resulting from synergistic effect of the hydration effect and hydrodynamic effect.

## Figures and Tables

**Figure 1 polymers-14-01917-f001:**
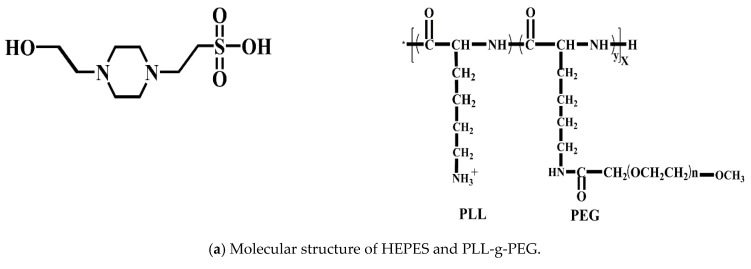
Schematic diagram of test methods and measuring devices: (**a**–**c**) are schematic diagrams of “grafting to” and “grafting from”; the "grafting to" polymer brush is fixed to the substrate surface by chemical bonding, and the "grafting from" polymer brush is achieved by surface-initiated polymerization. (**d**) is the friction pair of the friction tester used to measure the friction and wear between the ball and plates grafted by polymer brush; and (**e**) is the film thickness measurement device to obtain the optical interference image and the film thickness curve for studying the film-forming property induced by polymer brush.

**Figure 2 polymers-14-01917-f002:**
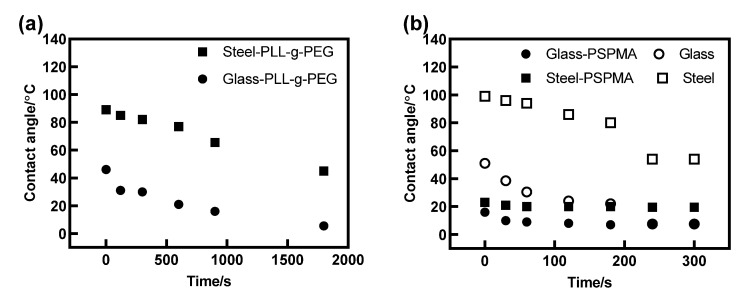
Contact angle of two polymers on glass and steel surfaces. (**a**) Surface lubricated by polymer PLL-g-PEG lubricating fluid; and (**b**) surface modified by polymer PSPMA brushes.

**Figure 3 polymers-14-01917-f003:**
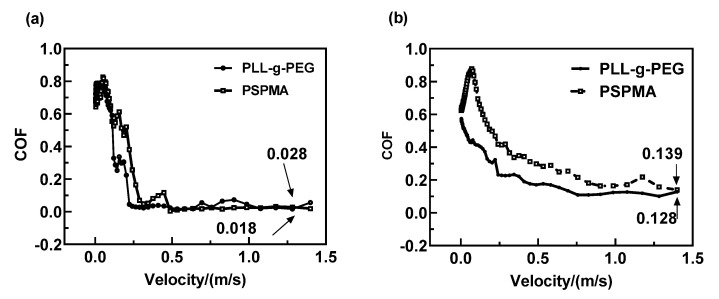
Comparison of Stribeck curves of two polymer brushes at 0.5 N and 3.5 N: (**a**) Load = 0.5 N; (**b**) Load = 3.5 N. With the increase in the entrainment speed, the surface modified by PSPMA had a lower coefficient of friction under 0.5 N load; the PLL−g−PEG modified surface had a lower coefficient of friction under 3.5 N load.

**Figure 4 polymers-14-01917-f004:**
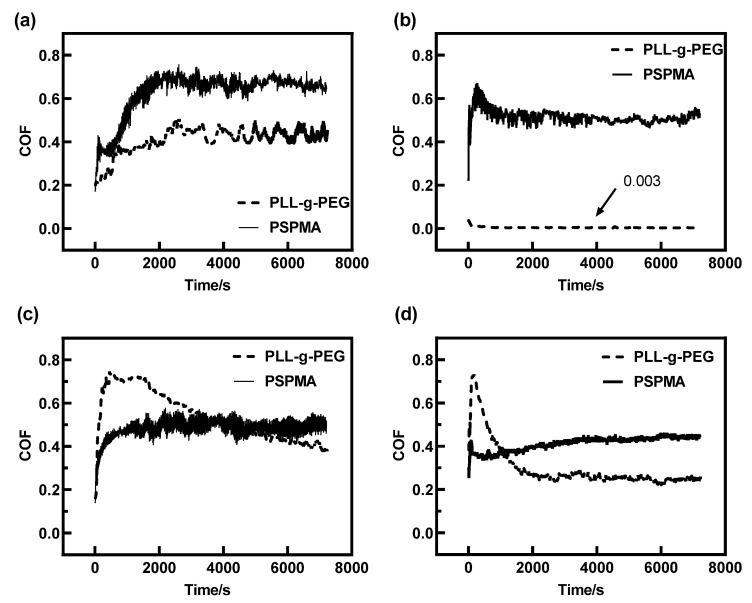
Comparison of friction curves of two polymer brushes vs. time at different loads and velocities: (**a**) Load = 0.5 N, U_e_ = 10 mm/s; (**b**) Load = 0.5 N, U_e_ = 300 mm/s; (**c**) Load = 3.5 N, U_e_ = 10 mm/s; and (**d**) Load = 3.5 N, U_e_ = 300 mm/s.

**Figure 5 polymers-14-01917-f005:**
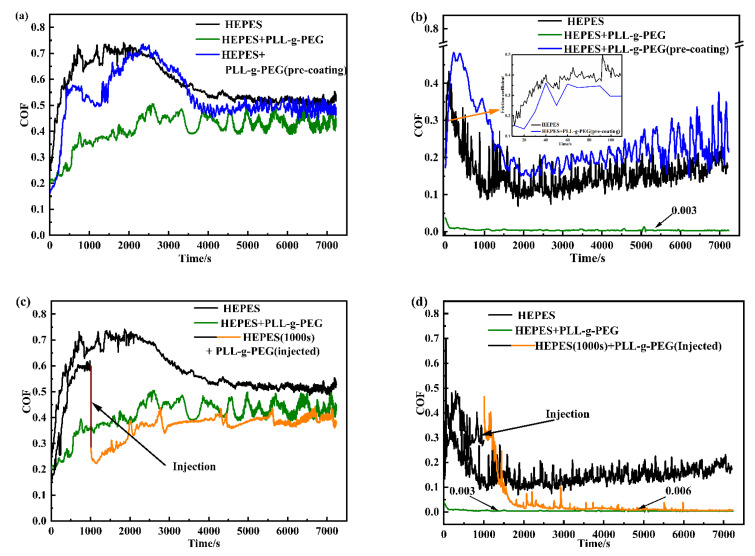
Curves of friction coefficients with time at 10 mm/s and 300 mm/s: (**a**) U_e_ = 10 mm/s; (**b**) U_e_ = 300 mm/s; (**c**) U_e_ = 10 mm/s; and (**d**) U_e_ = 300 mm/s.

**Figure 6 polymers-14-01917-f006:**
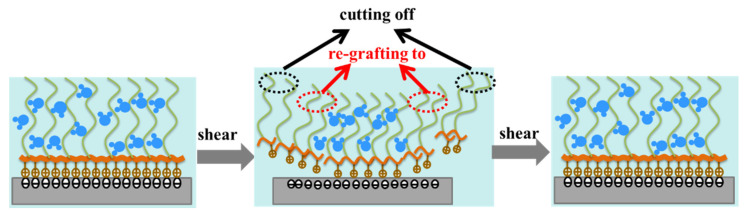
Schematic diagram of “self-healing” of PLL-g-PEG: the polymer was grafted to the surface (“grafting to”), sheared under shearing force (“cutting off”), then grafted to the surface (“re-grafting to”).

**Figure 7 polymers-14-01917-f007:**
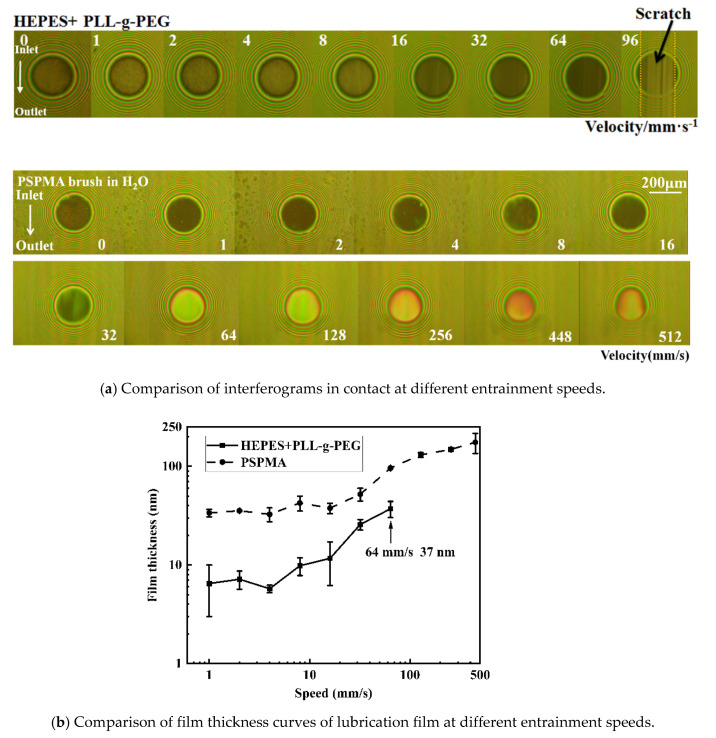
Comparison of interferograms and film thickness with entrainment speeds: from the light interferogram in (**a**), the film thickness value under the entrainment speed was measured, as shown in (**b**).

**Figure 8 polymers-14-01917-f008:**
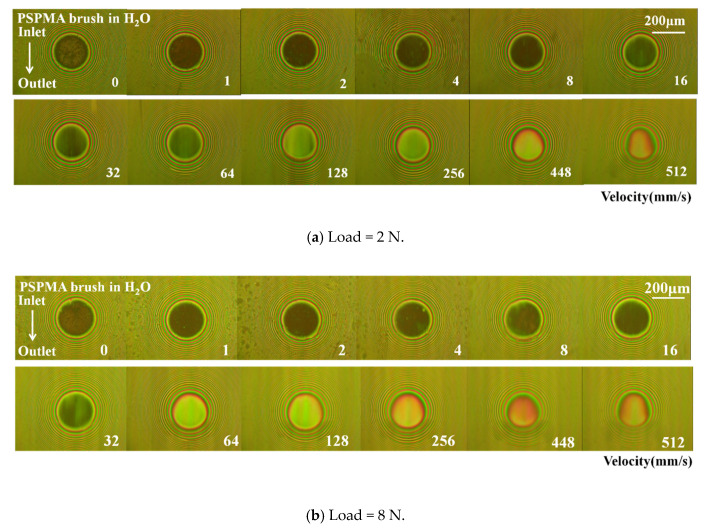
Optical interference images and film thickness curves of polymer brush PSPMA varying with entrainment speeds under different loads: (**a**–**d**) are the light interference diagrams of 2 N, 8 N, 16 N, and 32 N, respectively, and the film thickness values in (**e**) were obtained from the light interference diagrams.

**Figure 9 polymers-14-01917-f009:**
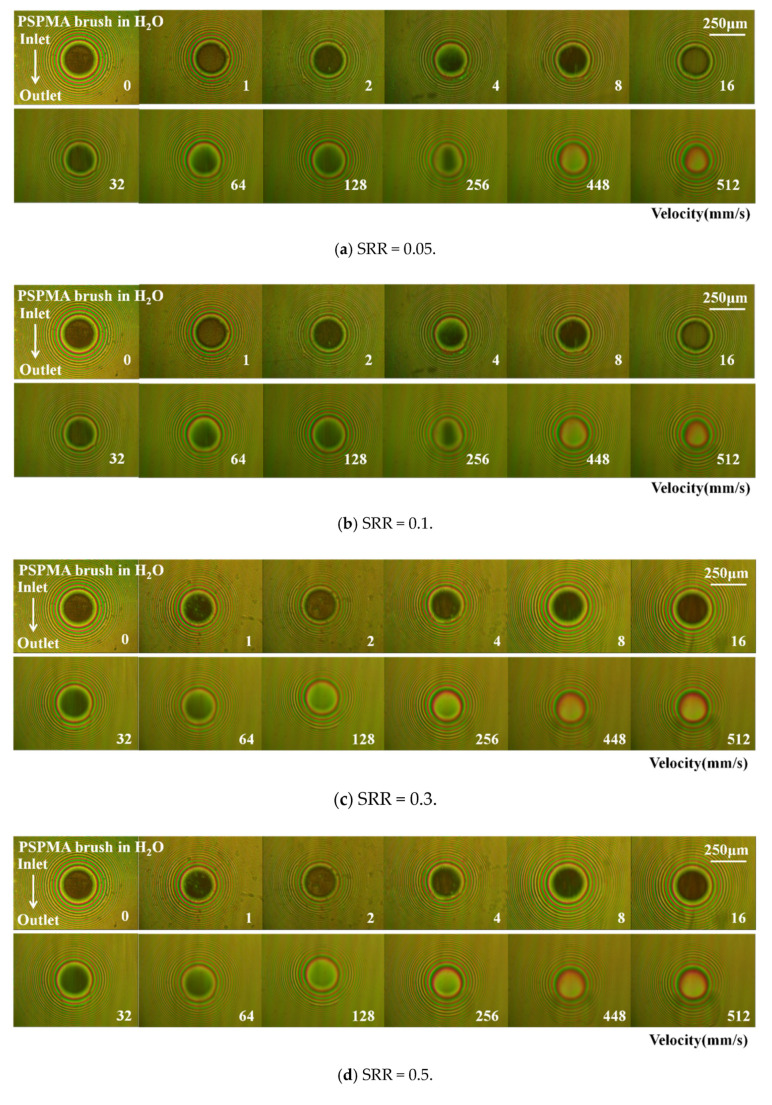
Optical interference images and film thickness curves of polymer brush PSPMA with different entrainment speeds under different slip-to-roll ratios: (**a**–**e**) is the light interference diagram for different slip ratios of SRR = 0.05, 0.1, 0.3, 0.5, and 0.7; (**f**) is the film thickness under the load of 4 N and different slip-roll ratios.

**Figure 10 polymers-14-01917-f010:**
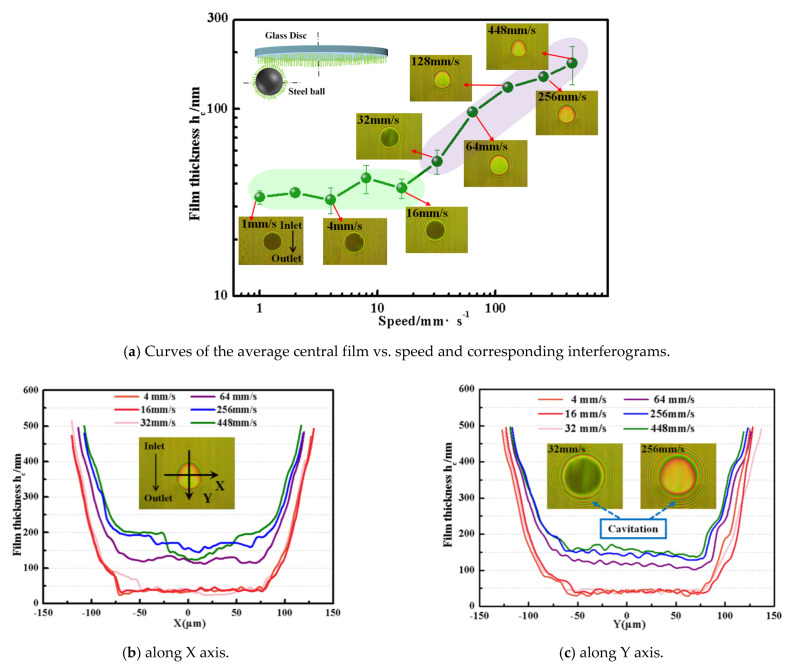
Film thickness curves and captured interferometric images in steel ball–glass disk contact grafted by polymer brush PSPMA at 4 N of load: (**a**) shows the film thickness values at different entrainment speeds as well as optical image; (**b**,**c**) show the film thickness values at different entrainment speeds along the X axis and Y axis.

## Data Availability

Not applicable.

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
