# Peer review of "Comparative Study on Macro-Tribological Properties of PLL-g-PEG and PSPMA Polymer Brushes"

_polymers, 2022, doi:10.3390/polym14091917_

Round 1

Reviewer 1 Report

Review of the manuscript "Comparative study on macro-tribological properties of PLL-gPEG and PSPMA polymer brushes". The idea of this paper seems interesting and the paper is reported in a quite clear way. I have only some minor comments:

  1. What was the hardness of the balls and plates used in tribological tests?
  2. On what basis the values of the tested parameters (load, velocity etc.) were selected? Please explain?
  3. All subfigures should have a description in the title of the figure, for example: Figure 2. Contact angle of two polymers on glass/steel surface: (a) surface lubricated by polymer PLL-g-PEG lubricating fluid, (b) surface modified by polymer PSPMA brushes.
  4. A deeper analysis could enrich the article. I recommend using SEM images taken on the surface of co-acting elements after tribological tests.
  5. It would be also worth clarifying where the results could be applied. Do the results have only cognitive or also applicable character?

Author Response

Point 1: What was the hardness of the balls and plates used in tribological tests?

 Response 1: The balls and plate/disc used in tribological tests was made of GCr15 steel and K9 glass respectively, the hardness of steel ball and glass plates/disc are 63~65HRC and 100HK, respectively. The relative description has been revised in the first paragraph in section 2.5 of manuscript.

Point 2: On what basis the values of the tested parameters (load, velocity etc.) were selected? Please explain?

Response 2: The test condition selected in this research was mainly to study the macro-tribological properties of polymer brushes. For the confirmation of the applied load, we refer to the test case in a paper titled “Self-healing behavior of a polyelectrolyte-based lubricant additive for aqueous lubrication of oxide materials” seen in Reference 9 of manuscript, where the Hertz contact pressure is 0.51GPa. Considering the different experimental materials used, the Hertz contact pressure was selected to be 217MPa and 416MPa, and the corresponding applied loads were 0.5N and 3N.

To select the velocity of 10 mm/s and 300 mm/s was based on Stribeck curve of polymer brush PLL-g-PEG at the load of 0.5N and 3.5N conducted on MCR302 as shown in Fig S1. The friction coefficient curve demonstrate that it was in BL lubrication regime at the velocity of 10 mm/s, while it was EHL lubrication regime at 300 mm/s, the purpose is to verify the effect of polymer brushes on tribological properties under different lubrication regime. The figure has been appended to Section Supplement.

 (a) Load = 0.5 N                       (b) Load = 3.5 N

Fig S1. Stribeck curves lubricated by HEPES and HEPES added by PLL-g-PEG

Point 3: All subfigures should have a description in the title of the figure.

Response 3: The authors are very grateful to the reviewers for this suggestion. All subfigures has been added by description in the title of the figure, please see Fig.1~ Fig.11 for details in modified manuscript.

Point 4: A deeper analysis could enrich the article. I recommend using SEM images taken on the surface of co-acting elements after tribological tests..

Response 4: The authors totally agree with the reviewer about a deeper analysis could enrich the article. Actually SEM images of steel balls grafted by PLL-g-PEG was got after the friction test at the load of 0.5 N with the magnifications of 150 and 800, respectively, which shows that PLL-g-PEG can also reduce the wear compared with pure HEPES. The authors also tried to further analyze the elements on worn scar using EDS, unfortunately, failed to find the difference by adding PLL-g-PEG into HEPES, which may be ascribed to very low content of polymer. The figure has been appended to the supplement section and named Fig S2.

Since PSPMA polymer brush had very weak effect in reduction friction and wear as mentioned in manuscript, further analyzed using SEM and EDS was not conducted.

Fig.S2 SEM micrographs of steel ball grafted PLL-g-PEG

Point 5: A deeper analysis could enrich the article. I recommend using SEM images taken on the surface of co-acting elements after tribological tests..

Response 5: The authors totally agree with the reviewer about the application of polymer brush, which has been described the last paragraph in Section 1. Introduction. According to the author's knowledge, polymer brush as coating has been successfully applied in urinary catheter due to its good lubrication performance, this greatly eases the pain of the patients. Polymer brush was also view as green lubrication additive. The authors believe that the results have cognitive and certain applicable character, for it was demonstrated in recent our research that polymer brush with higher hydration can achieve higher film thickness and load capacity by comparing two polymer brush with different hydration degree.   

Reviewer 2 Report

The work is focused on improving tribological properties of tribosystems through applying perspective polymer brushes on rubbing surfaces. The paper is relevant, discloses approaches to the methods of obtaining such coatings, lubricating mechanisms and tribological behavior. However, there are some comments that may help:

  1. The authors mention that polymer brushes improve tribological properties of some tribosystems working with water-based lubricants. I find it reasonable to address what kind of materials are applicable for polymer brushes (metals, ceramics, polymers, organics)? Are there any limitations or restrictions?
  2. The text is overloaded with too long sentences making it harder to understand.
  3. Figure 1(d) – check the spelling (Friction).
  4. Disclose the motivation for choosing materials for testing balls and plates (Section 2.5).
  5. It is not clear from Section 2, whether the glass and steel plates were used simultaneously in one rig or there were two or more experiments involving either steel plates and steel balls or glass plates and glass balls. What were the materials of the tribosystem in each experiment?
  6. What kind of the contact angle meter was used within the experiments (Section 2.5)?
  7. Section 3.1: It is preferred to add a sentence or two describing the importance of a contact angle and wettability to preserve cohesion of the text. In addition, why would authors conclude that PLL-g-PEG polymer completely grafted after 30 minutes while it is seen from Fig. 2a, that the contact angle is still decreasing. Was the experiment continued further? As far as it is understood, contact angle is taken as a criterion indicating obtaining the desired PLL-g-PEG polymer on glass.
  8. The first paragraph of the Section 3.2 is better to be placed in the Section 2 as it discloses friction tests techniques and parameters.
  9. Figures 3 and 4 are preferred in higher resolution.
  10. I would replace the term “home-made” with “laboratory set for.. “.
  11. What is the oil pool mentioned in the Section 3.4? Please, disclose.
  12. Did the authors managed to measure wear of both the plate and the ball after tests? This parameter is as important as friction coefficient.
